# Language-guided Task Adaptation for Imitation Learning

## Abstract

We introduce a novel setting, wherein an agent needs to learn a task from a demonstration of a related task with the difference between the tasks communicated in natural language. The proposed setting allows reusing demonstrations from other tasks, by providing low effort language descriptions, and can also be used to provide feedback to correct agent errors, which are both important desiderata for building intelligent agents that assist humans in daily tasks. To enable progress in this proposed setting, we create two benchmarks—Room Rearrangement and Room Navigation—that cover a diverse set of task adaptations. Further, we propose a framework that uses a transformer-based model to reason about the entities in the tasks and their relationships, to learn a policy for the target task.

## Introduction

Teaching learning agents how to perform a new task is a central problem in artificial intelligence. One approach is imitation learning (Argall et al., 2009) which involves showing demonstration(s) of the desired task to the agent, from which the agent can infer the demonstrator's intent and learn a policy for the task. However, for each new task, the agent must be given a new set of demonstrations, which can be burdensome as the number of tasks grow, since providing demonstrations is often a cumbersome process. On the other hand, techniques in instruction-following (MacMahon et al., 2006; Vogel & Jurafsky, 2010; Chen & Mooney, 2011) communicate the target task to a learning agent using language, which is a much more natural modality, particularly for non-experts. But as the complexity of tasks grows, providing intricate details using natural language can also become challenging.

This motivates a new paradigm that combines the strengths of both imitation learning and natural language. To this end, we propose a novel setting—given a demonstration of a task (the *source task*), we want an agent to complete a somewhat different task (the *target task*) in a **zero-shot** setting, that is, without access to *any* demonstrations for the target task. The difference between the source task and the target task is communicated using natural language. For example, consider an environment consisting of objects in a room, as shown in Figure 1. Suppose we have a robot, to which we have already provided the demonstration shown on the left, and it learns to perform that task. Now, we want to teach it to go to the opposite side of the table without providing a new demonstration. We posit that given a demonstration for the source task, and a linguistic description of the difference between the source and the target tasks, such as "Go to the opposite side of the wide table", the robot

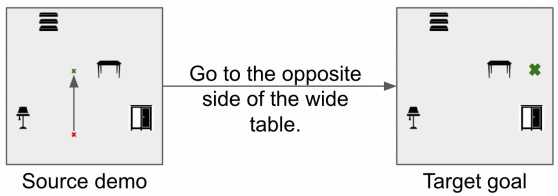

Figure 1: Example of the setting: The left image shows a demonstration of the source task, where the red point is the initial location of the agent, and the green point is the final location. The image on the right shows the target task, with the desired goal location marked with the green 'x'. The objective is to train an agent to perform the target task without any demonstrations of the target task, which requires reasoning about relative positions of entities.

should be able to infer the goal for the target task. Note that, to infer the target goal, neither the source demonstration, nor the description, is sufficient by itself, and the agent must therefore combine information from both the modalities.

This setting has several promising use cases. First, it allows reusing demonstrations from related tasks, requiring only low effort language, instead of additional demonstrations for new tasks. Second, for complex tasks, intricate details that are harder to communicate using language alone can be demonstrated, while small differences between related

tasks can be communicated using language. Third, it can be used for correcting the behavior of an agent if it makes errors, where the agent's current behavior can be seen as a demonstration, and a natural language correction can be provided to guide the agent towards the correct behavior.

We introduce two new benchmarks for our problem setting—Room Rearrangement and Room Navigation. These domains cover different types of environments, including discrete and continuous state and action spaces, and short and long horizon tasks. Further, they include several different types of natural language adaptations. We construct datasets with both template-based and natural language descriptions collected from actual humans.

Finally, we develop an approach to learn a policy for the target task in this new setting. Since language often describes modifications in terms of *entities* and their relationships, we propose a relational approach—RElational Task Adaption for Imitation with Language (RETAIL). The approach contains two independent components: (1) inferring a reward function for the target task, which is then used to learn a policy using reinforcement learning (RL), and (2) learning a policy for the source task, and adapting the policy for the target task. We show that combining these two components results in a robust policy learning framework for the proposed setting.

To summarize, this work makes the following contributions. First, we propose a new setting that involves learning a target task in a zero-shot manner, given the demonstration of a source task and a natural language description of the difference between the source and target tasks. Second, we create two benchmarks and construct datasets for each, which would enable exploring this new problem setting. Third, we propose the RETAIL framework, and demonstrate its success on these benchmarks.

## Related Work

**Imitation Learning.**    In imitation learning, the agent is provided with demonstration(s) of a task, and needs to infer the demonstrator's intent, thereby learning a policy to complete the task (Argall et al., 2009; Pomerleau, 1989; Ross et al., 2011; Abbeel & Ng, 2004; Ramachandran & Amir, 2007; Ziebart et al., 2008; Finn et al., 2016; Ho & Ermon, 2016; Fu et al., 2017). Our proposed setting differs from standard imitation learning, since the agent is provided with demonstration(s) of the source task, but needs to learn a policy for a related but different target task, the difference being communicated using language.

**Transfer Learning.**    Transfer learning, particularly in the context of reinforcement learning, involves an agent trained on a source task that needs to be adapted to a related but different target task. Various approaches have been developed for transfer learning in RL, which can be classified into different categories based on how the source and target tasks differ, and what information is being transferred (Taylor & Stone, 2009; Zhu et al., 2020). Our setting can be seen as an instance of transfer learning, where the transfer is guided using language.

**Language as Task Description.**    In a large class of approaches, which can broadly be termed as *instruction-following*, language is used to communicate the task to a learning agent, wherein, the agent is given a natural language command for a task, and is trained to take a sequence of actions that complete the task (Anderson et al., 2018; Fried et al., 2018; Wang et al., 2019; Tellex et al., 2011; Hemachandra et al., 2015; Arkin et al., 2017; Shridhar et al., 2020; Stepputtis et al., 2020; Misra et al., 2016; Sung et al., 2018). Our proposed setting is different from instruction-following, in that the goal of the target task is not communicated using language alone; instead, a demonstration for a related task (source task) is available, and language is used to communicate the difference between the two tasks. Thus, the information in the demonstration and the language complement each other.

**Language to Aid Learning.**    Several approaches have been proposed in the past that use language to aid the learning process of an agent. In a reinforcement learning setting, this could take the form of a language-based reward, in addition to the extrinsic reward from the environment (Luketina et al., 2019; Goyal et al., 2019; 2020b; Kaplan et al., 2017), or using language to communicate information about the environment to the agent (Wang & Narasimhan, 2021; Branavan et al., 2012; Narasimhan et al., 2018).

**Relational Reasoning.**    Relational reasoning involves representing the inputs of a learning system using a set of *entities*, and the model learns a task by reasoning about the relationships between these entities. Various techniques have been proposed for relational reasoning (Scarselli et al., 2008; Kipf & Welling, 2016; Velickovic et al., 2017; Goyal

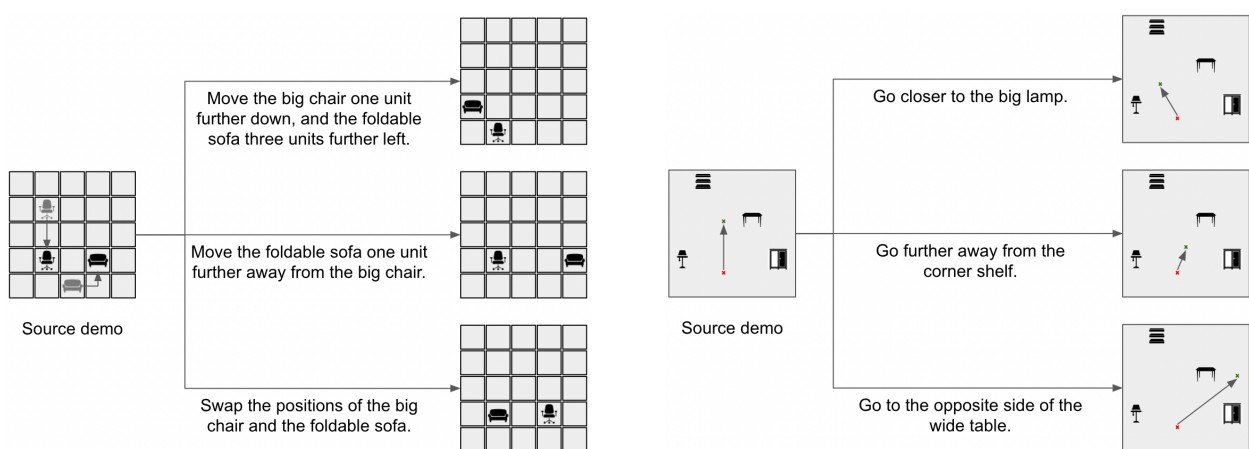

Figure 2: Adaptations used in the Room Rearrangement (left) and Room Navigation (right) domains.

et al., 2020a). These approaches have been shown to be effective for various machine learning problems, including reinforcement learning (Džeroski et al., 2001; Zambaldi et al., 2018; Zhou et al., 2022; Narasimhan et al., 2018), language grounding (Santoro et al., 2017; Dong et al., 2020), and modeling passive dynamics (Didolkar et al., 2021). In our work, we propose a relational model to use a source task demonstration to learn a target task, guided by language.

## Problem Definition

Consider a **goal-based task**, which can be defined as a task where the objective is to reach a designated goal state in as few steps as possible. It can be expressed using the standard Markov Decision Process (MDP) formalism, as $M = \langle S, A, P, g \rangle$, where $S$ is the set of all states, $A$ is the set of all actions, $P : S \times A \times S \to [0, 1]$ is the transition function, and $g \in S$ is the unique goal state. The reward function can take multiple different forms, for example, $R(s, s') = \mathbb{1}[s' = g]$, such that an agent following the optimal policy under the reward function reaches the goal state $g$ from any state $s \in S$, in as few steps as possible.

Let $\mathcal{T} = \{M_i\}_{i=1}^N$ be a **family** of goal-based tasks $M_i$, each with a distinct goal $g_i$ Kroemer et al. (2019). For instance, in the environment shown in Figure 1, a goal-based task consists of navigating to a goal state $g$, while $\mathcal{T}$ is the set of all such navigation tasks in the environment.

Let $T_{src}, T_{tgt} \in \mathcal{T}$ be two tasks, and $L$ be a natural language description of the difference between the tasks. Given a demonstration for the source task $T_{src}$, and the natural language description $L$, our objective is to train an agent to complete the target task $T_{tgt}$ in a **zero-shot setting**, i.e., without access to the reward function or demonstrations for the target task.

## Benchmark Datasets

We created two benchmark environments: Room Rearrangement and Room Navigation. For each environment, we construct a dataset of (source demonstration, language, target demonstration) triplets, where the demonstrations are generated using a planner, and the language is generated using templates. Further, we collect natural language descriptions for a subset of these datapoints, using Amazon Mechanical Turk (AMT). The details of the environments, the datasets, and natural language data collection are described below.

**Objects.** We use a common set of objects in both the environments, which we describe here. There are 6 distinct nouns—`Chair`, `Table`, `Sofa`, `Light`, `Shelf`, and `Wardrobe`. Further, each object can have one of 6 attributes—`Large`, `Wide`, `Wooden`, `Metallic`, `Corner`, and `Foldable`. The resulting 36 (attribute, noun) pairs are split into 24 train pairs, 6 validation, and 6 test pairs. Datapoints for each split use pairs only for that split. This ensures that the model will encounter unseen (attribute, noun) pairs during test time. The agent is also treated as an entity with both the attribute and the noun set to a special symbol `Agent`.

Table 1: Examples of template-generated and natural language descriptions collected using AMT.

| | Template | Natural language paraphrase |
|---|---|---|
| 1. | go further away from the metallic table | Increase your distance from the metallic table. |
| 2. | go closer to the foldable light | Move in the direction of the light that is foldable |
| 3. | go to the opposite side of the corner light | Move across from the corner light. |
| 4. | move the large chair one unit farther from the wide couch | Increment the distance of the big chair from the wide couch by one. |
| 5. | move corner table two units further left and metallic shelf one unit further backward | slide the corner table two units left and move the metal shelf a single unit back |
| 6. | move the large table to where the large sofa was moved, and vice versa | swap the place of the table with the sofa |

**Room Rearrangement Environment.** The Room Rearrangement Environment consists of a $5 \times 5$ grid, with 2 distinct objects. The objective is to move each object to a desired goal position. The agent and the objects are spawned randomly in the grid. The action space for the agent consists of 7 actions—Up, Down, Left, Right, Grasp, Release, and Stop. More details about the transition dynamics are provided in the Appendix.

**Room Navigation Environment.** The Room Navigation Environment consists of a 2D arena, $(x, y) \in [-100, 100]^2$, with 4 distinct objects. The agent is spawned at a random location in the arena, and needs to navigate to a desired goal position. The action space for the agent is $(dx, dy) \in [-1, 1]^2$. The episode terminates when the agent takes an action with an $\ell_2$-norm less than 0.1.

**Adaptations.** For each domain, we create three types of adaptations. For Room Rearrangement, these adaptations involve specifying an absolute change in the goal position of each entity, the relative change in the goal position of one entity with respect to the other, and swapping the goal positions of the entities. For Room Navigation, these adaptations involve moving closer to an entity, moving further away from an entity, and going to the opposite side of an entity. For each adaptation, we create a template to generate linguistic descriptions. See Figure 2 for examples of these adaptations and corresponding linguistic descriptions.

Together, these environments cover various types of adaptations, such as specifying modifications to one versus several entities, providing absolute modifications to an entity's position (e.g., "move the table one unit further left") versus modifications that are relative to other entities (e.g., "move the table one unit away from the sofa"). Further, these domains cover different types of MDPs, with Room Rearrangement being a discrete state and action space environment, with a relatively short horizon, while Room Navigation being a continuous state and action space environment, with a longer horizon. [1] Finally, the Room Navigation domain has a unique optimal path (i.e. a straight line path between the initial state and the goal state), while the Room Rearrangement domain admits multiple optimal paths (e.g. if reaching an entity requires taking 2 steps to the right and 1 step upwards, these steps can be performed in any order). Thus, these two domains make a robust testbed for developing techniques for the proposed problem setting.

**Language Data.** For each pair of source and target tasks in the dataset, we start by generating linguistic descriptions using templates, such as, "Move attribute1 obj1 one unit closer to the attribute2 obj2". For each type of adaptation described above, we create one template, resulting in 6 adaptation templates across both the benchmarks. We ensure that for all these templates, the target task cannot be inferred from the description alone, and thus, the model must use both the demonstration of the source task and the linguistic description to infer the goal for the target task. Next, we crowdsourced natural language for a subset of these synthetic (i.e. template-generated) descriptions using AMT. Workers were provided with the synthetic descriptions, and were asked to paraphrase these descriptions. Some examples of template-generated and natural language descriptions are shown in Table 1.

---

[1]On average, an optimal policy completes a task in the Room Rearrangement domain in about 30 steps, while in the Room Navigation domain in about 150 steps.

**Datasets.** For each adaptation template, 5,000 datapoints were generated for training, 100 for validation of the reward and goal learning, 5 for tuning the RL hyperparameters, and 10 for the RL test set. This gave us (1) a training dataset with 15,000 datapoints for each benchmark, (2) a validation dataset for supervised learning with 300 datapoints, (3) a validation set for RL with 15 datapoints, and (4) a test set for RL with 30 datapoints. We collect natural language paraphrases for 10% of the training datapoints, and all the datapoints in the other splits.

Since people often describe objects/tasks in the real-world in reference to other objects, we designed our adaptations such that most of them are relational in nature, that is, require reasoning about relative positions of entities. Consequently, we propose a relational approach for our setting, which we describe next.

## RElational Task Adaptation for Imitation with Language (RETAIL)

We propose the RElational Task Adaptation for Imitation with Language (RETAIL) framework that takes in a source demonstration, $\tau_{src}$, and the difference between the source and target tasks described using natural language, $l$, to learn a policy for the target task $\pi_{tgt}$. The framework consists of two independent approaches, as shown in Figure 3. The first approach—Relational Reward Adaptation—involves inferring a reward function for the target task $R_{tgt}$ using the source demonstration $\tau_{src}$ and language $l$, from which a policy for the target task $\pi_{tgt}$ is learned using RL. The second approach—Relational Policy Adaptation—involves learning a policy for the source task $\pi_{src}$ from the source demonstration $\tau_{src}$, which is then adapted using language $l$ to obtain a policy for the target task $\pi_{tgt}$.

For both these approaches, we assume access to a training set $\mathcal{D} = \{(\tau_{src}^i, \tau_{tgt}^i, l^i)\}_{i=1}^N$, where for the $i^{th}$ datapoint, $\tau_{src}^i$ is a demonstration for the source task, $\tau_{tgt}^i$ is a demonstration for the target task, and $l^i$ is the linguistic description of the difference between the source task and the target task.

We propose a relational model since many adaptations require reasoning about the relation between entities (e.g. "Move the big table two units away from the wooden chair"). Since entity extraction is not the focus of this work, we assume access to a set of entities for each task, where each entity is represented using two one-hot vectors, corresponding to an attribute and a noun. The details of attributes and nouns used in our experiments have been described in the previous section. Further, each state is represented as a list, where element $i$ corresponds to the $(x, y)$ coordinates of the $i^{th}$ entity. Finally, we assume that the number of entities, denoted as $N_{entities}$, is fixed for a given domain.

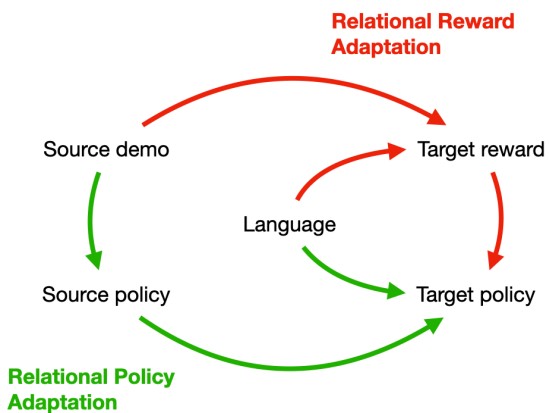

Figure 3: We present two independent approaches to learn a target task policy – relational reward adaptation and relational policy adaptation. Finally, we show how to combine these two approaches.

We start by describing some common components used in both the approaches.

**Entity Encoder.** We assume each entity is represented using two one-hot vectors, corresponding to an attribute and a noun. These two encodings are passed through embedding layers for attributes and nouns respectively, to obtain dense vector representations. Further, each entity's position in a state is represented using (x, y) coordinates. These coordinates are passed through a linear layer to obtain a dense vector representation. The attribute, noun, and position dense vectors are concatenated to get the final vector representation of the entity, $e_i$.

**Language Encoders.** We experiment with 4 different ways of encoding language. First, we use a pretrained CLIP model (Radford et al., 2021), which has been shown to be effective at language grounding tasks, to obtain an embedding for each token in the description. The parameters of the pretrained model are kept frozen during the training of the downstream network. Second, instead of a pretrained CLIP model, we use a pretrained BERT model (base, uncased; (Devlin et al., 2018)). As before, the pretrained model is kept frozen. Third, instead of using a pretrained BERT model, we experiment with a randomly initialized BERT model that is learned along with the downstream network. Finally,

we use GloVe word embeddings (Pennington et al., 2014) followed by a two-layer bidirectional LSTM (Hochreiter & Schmidhuber, 1997). The GloVe+LSTM and randomly initialized BERT models are more flexible, allowing them to learn representations for words and sentences that are specialized for the task at hand, while the pretrained CLIP and BERT models can potentially leverage the external knowledge seen during the pretraining phase. While pretrained BERT encodes language independent of its grounding, the CLIP model is pretrained on multimodal data, which is likely more useful for our setting which requires language grounding.

Next, we describe the details of Relational Reward Adaptation and Relational Policy Adaptation.

### Relational Reward Adaptation

We define the reward $R(s, s')$ using a potential function as, $R(s, s') = \phi(s') - \phi(s)$. Thus, the problem of reward learning is reduced to the problem of learning the potential function $\phi(s)$. We decompose the potential function learning problem into two subproblems: (1) predicting the goal state for the target task given the source goal and the language, $g_{tgt} = Adapt(g_{src}, l)$, and (2) learning a distance function between two states, $d(s, s')$. The potential function for the target task is then defined as follows:

$$\phi_{tgt}(s|g_{src}, l) = -d(s, Adapt(g_{src}, l))$$

### Goal Prediction

Given a set of entities $E$, a goal state for the source task represented as a list of positions of each entity ($g_{src}$), and a natural language description of the difference between the tasks ($l$), the goal predictor network is trained to predict the goal position for each entity in the target task ($g_{tgt}$).

First, the entities in the goal state for the source task, and the language description are encoded using the Entity Encoder and Language Encoder module described above. The encoded goal states for the source task, and the token embeddings generated by the language encoder are concatenated to create a single sequence of tokens, which is passed through a transformer layer. The first $N_{entities}$ tokens of the output sequence are projected back to the $\mathbb{R}^2$ space using a multi-layer perceptron with three linear layers and ReLU non-linearities between them to get the predicted goal state of each of the $N_{entities}$ entities under the target task.

A diagram of the goal predictor neural network is shown in Figure 4.

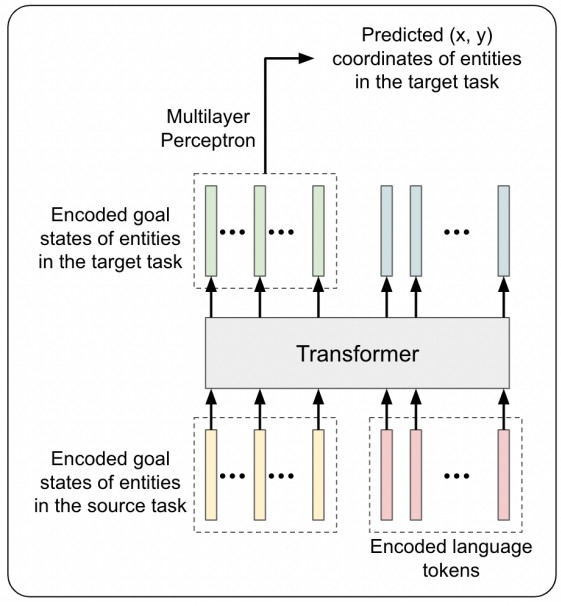

Figure 4: Neural Network architecture for relational goal prediction.

### Distance Function Learning

The distance function takes in two states $s$ and $s'$, and predicts the distance between them. While an $\ell_2$-distance between the states can be directly computed, this may not be ideal in many domains. Therefore, we learn a neural network $\psi$ with two linear layers and a ReLU non-linearity between them to project the raw states into an embedding space. The distance between the states $s$ and $s'$ is then computed as $d(s, s') = \|\psi(s) - \psi(s')\|_2$.

### Training

To train the model, we assume access to a dataset $\mathcal{D} = \{(\tau_{src}^i, \tau_{tgt}^i, l^i)\}_{i=1}^N$.

The goal prediction module is trained by using the final states in the source and target demonstrations, as the source and target goals respectively. We minimize the mean absolute error between the gold target goal state, $g_{tgt}$ and the

predicted target goal state, $\hat{g}_{tgt}$:

$$L_{goal} = \frac{1}{N} \sum_{i=1}^{N} \|g_{tgt} - \hat{g}_{tgt}\|_1$$

To train the distance function, two states $s_i$ and $s_j$ are sampled from a demonstration $\tau$, which can be the source or the target demonstration for the task, such that $i < j$. The model is trained to predict distances such that $d(g, s_i) > d(g, s_j)$, where $g$ is the goal state for the demonstration. This is achieved using the following loss function:

$$L_{dist} = -\sum_{s_i, s_j, g} \log\left(\frac{\exp(d(g, s_i))}{\exp(d(g, s_i)) + \exp(d(g, s_j))}\right)$$

This loss function has been shown to be effective at learning functions that satisfy pairwise inequality constraints (Christiano et al., 2017; Brown et al., 2019).

The goal prediction and distance function modules are independently trained using the dataset $\mathcal{D}$. We used an Adam optimizer (Kingma & Ba, 2015) to train the networks for 100 epochs each. A validation set was used to tune hyperparameters via random search.

The learned goal prediction and distance function modules are combined to obtain a reward function for the target task, which is then used to train a policy using reinforcement learning. More details about this step are provided in the Experiments section.

## Relational Policy Adaptation

Instead of learning a model to infer the reward function for the target task from the source demonstration and language, in this section, we describe an alternate approach wherein we learn a model to infer the target task policy from the source task policy.

First, a goal-conditioned policy $\pi(a|s, g)$ is learned using all the source and target demonstrations—given the goal state for a task, $g$, (which is assumed to be the last state in the demonstration), and another state, $s$, we use behavior cloning to learn a policy that predicts the action to be taken at state $s$. We use a neural network to parameterize this policy, wherein the states $g$ and $s$ are concatenated and then passed through a multi-layer perceptron to predict the action at state $s$.

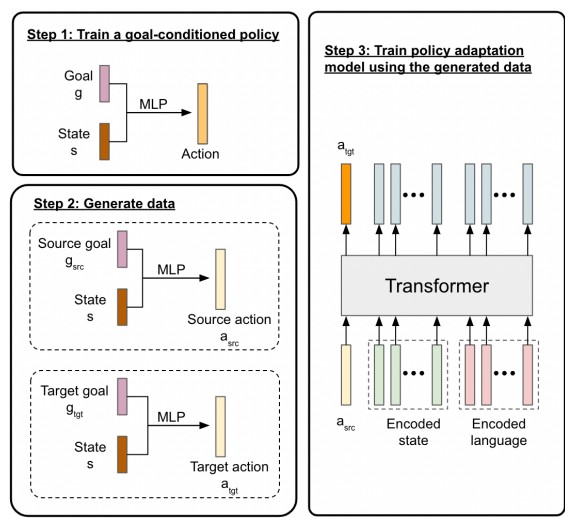

Figure 5: Relational Policy Adaptation approach

The learned model is then used to generate data of the form (state, language, source action, target action). For each datapoint of the form $(\tau^i_{src}, \tau^i_{tgt}, l^i)$ in the original dataset, the states in the source and target demonstrations are passed through the learned goal-conditioned policy, passing in the source task goal and the target task goal to obtain the actions in the source and target tasks respectively:

$$a_{src} \sim \pi(a|s, g_{src}) \ ; \quad a_{tgt} \sim \pi(a|s, g_{tgt})$$

This data is used to train a transformer-based adaptation model, that takes in the source action, the entities in the state $s$, and the language to predict the target action. The entities and language are encoded using the Entity Encoder and Language Encoder described above. See Figure 5 for a diagram of the approach.

During evaluation, we are given the source demonstration and language, as before. We use the goal-conditioned policy $\pi(a|s, g)$ to first predict the action for the current state under the source task, and then pass this predicted action, along with the encoded entities and language to the adaptation model, to obtain the action under the target task. This action is then executed in the environment. The process is repeated until the STOP action is executed or the maximum episode length is reached.

Note that this approach does not involve reinforcement learning to learn the policy.

**Combining Reward and Policy Adaptation**

So far, we've described the relational reward adaptation approach that infers a reward function for the target task, and the relational policy adaptation approach that infers an initial policy for the target task. In this section, we describe how these approaches can be combined.

Recall that the actor-critic model in the PPO algorithm consists of a policy network and a value network. We use the output of the policy adaptation and the reward adaptation approaches to initialize these networks respectively.

Note that the architectures for the policy and value networks in PPO are different from the architectures of the networks in policy and reward adaptations. Specifically, the networks in PPO are trained for a single target task, and therefore only take the state as input, whereas the policy and reward adaptation approaches are shared across different tasks and are therefore conditioned on language as well. As such, we cannot directly initialize network weights in PPO, and therefore use knowledge distillation (Hinton et al., 2015) to initialize the networks.

Thus, our full approach can be described as follows (see Figure 6):

1. Train the reward adaptation and policy adaptation models using supervised learning independently, as detailed in the previous sections.

2. Use knowledge distillation to initialize the value network for PPO, updating the PPO value network towards the potential predicted by the reward adaptation approach.

3. Use knowledge distillation to initialize the policy network for PPO, updating the PPO policy network towards the action probabilities predicted by the policy adaptation approach for the target task.

4. Finetune the action and value networks using PPO with the rewards predicted by the reward adaptation approach.

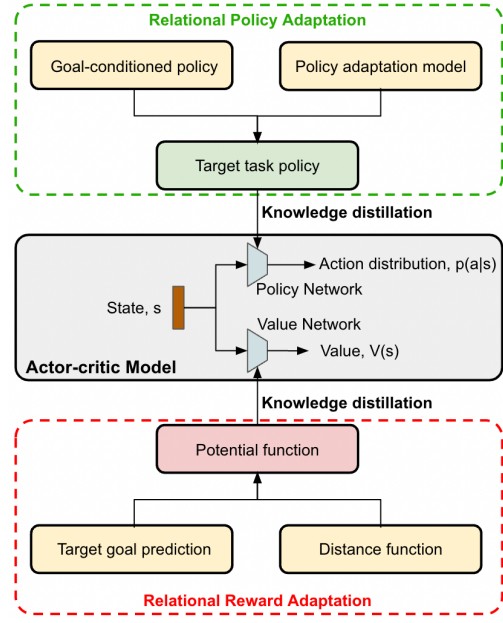

Figure 6: The combined approach

For knowledge distillation, states from the demonstration data are sampled uniformly at random.

Importantly, we found that the action network initialized using knowledge distillation usually has a low entropy, and therefore finetuning it directly does not result in good performance. To ameliorate this issue, the entropy of the action network must be kept sufficiently high for it to still allow some exploration. In the continuous control case, we achieve this by increasing the standard deviation of the action network, tuned using the validation set. In the discrete domain, since there is no explicit parameter to control the entropy in the action network, the knowledge distillation step has an additional loss term to penalize low-entropy solutions.

## Experiments

### Relational Reward Adaptation

**Policy Training.**   The goal prediction and distance function learned on the training set are used to train a policy for each task in the test set with the hyperparameters found to be optimal for the 5 validation RL tasks. We report the total number of successful episodes at the end of 500,000 and 100,000 timesteps respectively for these domains, averaged over three RL training runs per target task. We use PPO as the RL algorithm for all our experiments (Schulman et al.,

Table 2: Success rates for different models on Room Rearrangement and Room Navigation domains. We report both the raw success rates (unnormalized), and success rates normalized by the oracle setting performance.

| Setting | No. of successes | | | |
|---|---|---|---|---|
| | Rearrangement | | Navigation | |
| | Unnormalized | Normalized | Unnormalized | Normalized |
| Reward Adaptation | $2996.02 \pm 136.21$ | $68.05 \pm 3.09$ | $247.98 \pm 20.51$ | $73.54 \pm 6.08$ |
| Oracle | $4402.78 \pm 410.67$ | $100.00 \pm 9.33$ | $337.22 \pm 7.34$ | $100.00 \pm 2.18$ |
| Zero reward | $121.02 \pm 4.25$ | $2.75 \pm 0.10$ | $0.29 \pm 0.04$ | $0.09 \pm 0.01$ |
| True goal, predicted distance | $4164.80 \pm 337.83$ | $94.59 \pm 7.67$ | $362.13 \pm 12.18$ | $107.39 \pm 3.61$ |
| Predicted goal, true distance | $3706.80 \pm 200.46$ | $84.19 \pm 4.55$ | $196.49 \pm 12.97$ | $58.27 \pm 3.85$ |
| Synthetic language | $3827.64 \pm 141.79$ | $86.94 \pm 3.22$ | $317.11 \pm 49.26$ | $94.04 \pm 14.61$ |
| Non-relational goal prediction | $869.89 \pm 115.12$ | $19.76 \pm 2.61$ | $0.38 \pm 0.17$ | $0.11 \pm 0.05$ |
| Combined approach | $8516.78 \pm 894.35$ | $193.44 \pm 20.31$ | $430.80 \pm 5.08$ | $127.75 \pm 1.51$ |

2017; Raffin et al., 2021). For the Room Rearrangement domain, the agent and the entities are initialized uniformly at random anywhere on the grid at the beginning of each episode. For the Room Navigation domain, the entities are initialized in the same positions as in the source task, while the agent is initialized uniformly at random in the arena.

**Evaluation Metrics.**   In the Room Rearrangement domain, an episode is deemed successful if both the entities are in the desired goal locations when the agent executes `Stop`, while for the Room Navigation domain, an episode is deemed successful if the $\ell_2$-distance between the agent's final position and the desired goal position is less than 5 units. (Recall that the total arena size is $200 \times 200$ units, and the episode ends when the agent executes an action with an $\ell_2$-norm less than 0.1 unit.)

## Results

In this section, we describe the performance of our full model, along with various ablations. Our results are summarized in Table 2. For each experiment, we run policy training with 5 random seeds, and report the mean and standard deviation of the number of successful episodes. Unless stated otherwise, we use the full set of synthetic and natural language descriptions for supervised training, and only natural language descriptions during testing for RL.

The first row corresponds to our full reward adaptation model, that learns a relational goal prediction model, and a distance function, which are then combined to form a reward function for RL. The target tasks are trained with natural language descriptions collected using AMT. The next two rows serve as approximate upper and lower bounds respectively. The second row corresponds to an oracle setting, wherein, a policy is trained with the true goal state for the target task, and the potential of a state is defined as the distance between the current state and the goal state. We use the $\ell_1$-distance for Rearrangement, and the $\ell_2$-distance for Navigation. For the third row, we define the reward function to be uniformly zero, and this result tells us how well a random policy would do on our target tasks.

We can observe that the proposed model is substantially better than the lower bound, but is about 70% as good as the oracle. As such, there is quite a bit of room for improvement to achieve performance close to the oracle.

Of the language encoders we used, we did not find any substantial difference between different encoders. This is likely because the language descriptions used in our experiments were relatively simple. However, as the proposed setting is extended to richer tasks, we expect these language encoders to perform differently.

To understand the effect of various design choices, we ran several ablation experiments, which we describe next. Since our full model consists of two learned components, the goal prediction module, and the distance function, we first study the impact of each of these components independently. We experiment with the following two settings: (1) the true target goal state, with the learned distance function (Row 4), and (2) the learned target goal prediction, with the true distance function, where the true distance function is as used in the oracle setting for each domain (Row 5). As expected, the distance function is easy to learn in these domains, and using the learned distance function instead of the true distance function leads to a small or no drop in performance. Most the performance drop comes from the goal prediction module, and therefore future modeling innovations should focus on improving the goal prediction module.

Next, we look at the performance difference between synthetic and natural language. Row 6 in Table 2 shows the number of successful episodes when using synthetic language only, both during training the goal prediction model,

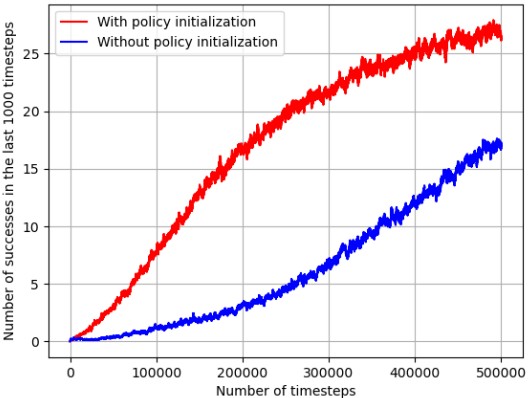 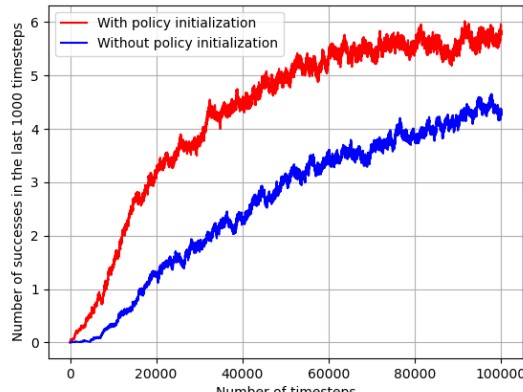

Figure 7: Learning curves comparing the policy training on target tasks when using uninitialized PPO networks and PPO networks initialized using policy adaptation, on the Rearrangement (left) and Navigation (right) domains.

and for learning the target task policy using RL during testing. In both the domains, using synthetic language is significantly better than using natural language, and is comparable to the oracle.

In order to analyze the benefit of using the relational model, we compare our approach against a non-relational model. Row 7 shows the results when using a non-relational model, where we use a multilayered perceptron with three linear layers, that takes in the entity vectors, goal positions of all entities in the source task, and the CLIP embedding of the final token in the description, all concatenated together as a single input vector, and outputs the goal positions of all entities in the target task as a single vector. This model is significantly worse than the relational model on both the domains, highlighting the benefit of using a relational approach for these tasks.

**Relational Policy Adaptation**

To evaluate this approach, we generate 100 rollouts using the trained models for each test task, and compute the number of successful episodes. For each rollout, we randomize the initial state as in the policy training experiments for Relational Reward Adaptation.

On the Rearrangement domain, the approach completes 15.33% tasks when using natural language, and 29.13% tasks when using synthetic language. On the Navigation domain, the approach results in 3.87% and 21.71% success when using natural and synthetic language respectively.

**Combining Reward and Policy Adaptation**

For the experiments here, we use synthetic and natural language for supervised learning, and only natural language for RL during evaluation. We report the number of successes when PPO is initialized randomly, as in Relational Reward Adaptation experiments, and when it is initialized using the adapted policy in the final row of Table 2. Further, Figure 7 shows the learning curves for these experiments.

We observe that on both the domains, initializing the policy network using the Relational Policy Adaptation approach and the value network using the Relational Reward Adaptation approach leads to a substantially faster policy learning on the target tasks, compared to randomly initialized PPO networks.

Some qualitative results using the reward and policy adaptation approaches are included in the appendix.

**Key Takeaways**

To summarize, our experiments demonstrate that: (1) Relational Reward Adaptation leads to successfully learning the target task from the source demonstration and language in many test tasks, but there is room for improvement;

(2) Relational Policy Adaptation can be used to complete some target tasks without RL, but there is a significant room for improvement; and (3) combining the two approaches followed by finetuning with RL leads to a much better performance than using either approach independently.

## Conclusions

We introduced a new problem setting, wherein an agent needs to learn a policy for a target task, given the demonstration of a source task and a linguistic description of the difference between the source and the target tasks, and created two relational benchmarks – Room Rearrangement and Room Navigation – for this setting. We presented two relational approaches for the problem setting. The first approach – relational reward adaptation – learns a transformer-based model that predicts the goal state for the target task, and learns a distance function between two states. These trained modules are then combined to obtain a reward function for the target task, which is used to learn a policy using RL. The second approach – relational policy adaptation – learns a transformer-based model that takes in a state, and the action at this state under the source task, to output the action at this state under the target task, conditioned on the source task goal and language. We show that combining these approaches results in effective policy learning.

We believe that the problem setting, benchmarks, and approaches presented here would enable further research along this line of work, including short-term directions such as combining the proposed approaches with entity extraction methods and handling variable number of entities, and long-term directions such as more realistic domains and tasks.

The problem setting would enable teaching multiple related tasks to an agent by providing a few demonstrations with multiple low-effort linguistic descriptions, thus reducing the burden of providing demonstrations on the end user. Further, as mentioned in the introduction, the problem setting is closely related to that of learning from feedback. As such, it may be informative to explore how the proposed approaches can be used for feedback, or vice versa.

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

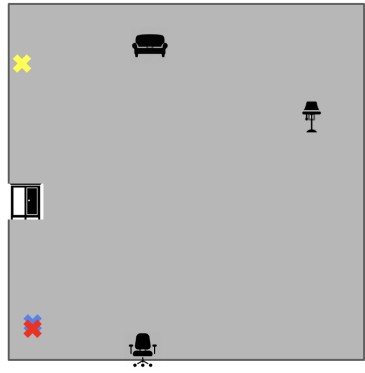
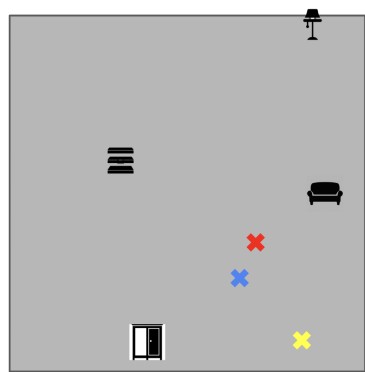

turn around to the other side
of the folding wardrobe

approached the big shelf

Figure 8: Visualization of predicted goal for two test datapoints using the reward adaptation approach. The yellow X denotes the goal position under the source task, and the red and blue X's denote the predicted and true goal positions under the target task.

## Appendix

### Dataset details

**Transition Dynamics for the Room Rearrangement Domain.** If the agent is on a cell that contains another object, the `Grasp` action picks up the object, otherwise it leads to no change. A grasped object moves with the agent, until the `Release` action is executed. The `Up`, `Down`, `Left`, and `Right` actions move the agent (and the grasped object, if any) by one unit in the corresponding direction, except when the action would result in the agent going outside the grid, or the two objects on the same grid cell. In these cases, the action doesn't result in any change. The `Stop` action terminates the episode.

## Qualitative Results

In this section, we report some qualitative results on the Navigation domain with reward and policy adaptation approaches.

In Figure 8, we show two examples of goal prediction using the Relational Reward Adaptation approach. In the first example, the predicted goal state is quite close to the true goal state under the target task, suggesting that the model is able to successfully recover the target task. In the second example, the predicted goal is somewhat farther from the true goal. A plausible explanation is that the model was not able to disambiguate the entity being referred to by language, and therefore computes the target goal position as a linear combination of distances to multiple entities.

In Figure 9, we show three examples of paths followed by the agent when following the actions predicted by the Relational Policy Adaptation approach (without any finetuning). In the first example, we see that the agent successfully reaches and stops at the true goal position under the target task. In the other two examples, we see that the agent gets somewhat close to the goal position under the target task, but doesn't actually reach it (and is also going towards the goal position under the source task). The errors seem to get larger as agent gets closer to the target goal, motivating a modified training algorithm wherein datapoints could be weighted differently based on how close the agent is to the goal position. We leave this investigation for future work.

## Compute Infrastructure

All experiments were run on a machine with 4 Quadro RTX 6000 GPUs, 64 CPUs, and 512 GB of RAM. We used the PyTorch deep learning framework (version 1.10.1) (Paszke et al., 2019), along with Stable Baselines 3 for RL (Raffin et al., 2021).

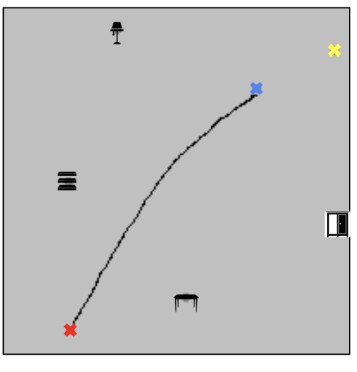 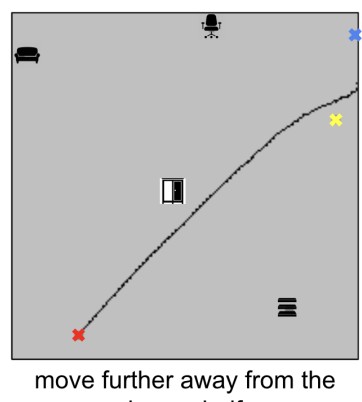 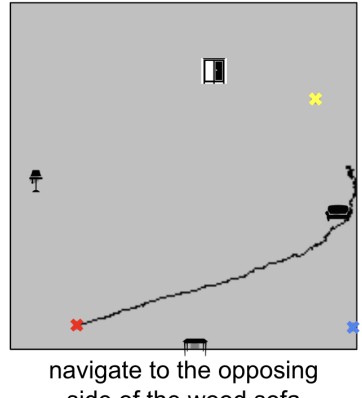

| approach the large rack | move further away from the large shelf | navigate to the opposing side of the wood sofa |

Figure 9: Visualization of predicted goal for three test datapoints using the policy adaptation approach. The red X denotes the initial position of the agent, the yellow X denotes the true goal position under the source task, and the blue X denotes the true goal position under the target task.

Table 3: Hyperparameters for target goal prediction in relational reward adaptation

| Hyperparameter | Values tried | Best value | |
| --- | --- | --- | --- |
| | | Rearrangement | Navigation |
| Batch size | 8, 16, 32, 64 | 16 | 16 |
| Learning rate | 1e-3, 1e-4, 1e-5 | 1e-4 | 1e-4 |
| Dimension of the space in which the noun and the attribute of an entity are projected | 8, 16, 32 | 32 | 32 |
| Hidden layer size for the MLP used after the transformer module | 32. 64. 128. 256. 512 | 128 | 128 |
| No. of heads in the transformer module | 1, 2, 3, 4, 5, 6, 7, 8 | 8 | 8 |
| No. of encoder layers in the transformer module | 1, 2, 3, 4 | 4 | 4 |
| Dimension of the feedforward layer in the transformer module | 32, 64, 128. 256 | 256 | 256 |
| Dropout probability in the transformer module | 0.1, 0.2, 0.3, 0.4 | 0.4 | 0.4 |

## Hyperparameters

Here, we describe the hyperparameters used in our approaches, along with the values tried. For each module, we used 8 random seeds to choose the value of each hyperparameter, and selected the best set of values using a validation set.

Table 4: Hyperparameters for distance learning in relational reward adaptation

| Hyperparameter | Values tried | Best value | |
| --- | --- | --- | --- |
| | | Rearrangement | Navigation |
| Batch size | 8, 16, 32, 64, 128, 256, 512 | 512 | 32 |
| Learning rate | 1e-3, 3e-4, 1e-3, 3e-3 | 3e-3 | 1e-3 |
| Hidden layer size for the MLP | 8, 16, 32, 64, 128, 256, 512 | 512 | 128 |

Table 5: Hyperparameters for RL using PPO

| Hyperparameter | Values tried | Best value | |
| --- | --- | --- | --- |
| | | Rearrangement | Navigation |
| Batch size | 4, 8, 16, 32, 64, 128, 256, 512, 1024 | 32 | 16 |
| Learning rate | 1e-6, 3e-6, 1e-5, 3e-5, 1e-4, 3e-4, 1e-3, 3e-3, 1e-2, 3e-2, 1e-1 | 3e-5 | 3e-4 |
| No. of steps per update | 8, 12, 16, 24, 32, 48, 64, 96, 128, 192, 256, 384, 512, 768, 1024, 1536, 2048 | 512 | 256 |
| No. of epochs | 1, 2, 5, 10, 20, 30, 50 | 50 | 10 |
| gae_lambda | 0.99, 0.98, 0.95, 0.9, 0.8 | 0.99 | 0.99 |
| clip_range | 0.1, 0.2, 0.3 | 0.3 | 0.2 |
| Entropy coefficient | 1e-5, 3e-5, 1e-4, 3e-4, 1e-3, 3e-3, 1e-2, 3e-2, 1e-1, 3e-1, 1 | 0.1 | 3e-4 |
| Max gradient norm | 0.05, 0.1, 0.2, 0.5, 1.0, 2.0, 5.0 | 2.0 | 1.0 |

Table 6: Hyperparameters for goal-conditioned policy learning in relational policy adaptation

| Hyperparameter | Values tried | Best value | |
| --- | --- | --- | --- |
| | | Rearrangement | Navigation |
| Batch size | 8, 16, 32, 64 | 64 | 32 |
| Learning rate | 1e-4, 3e-4, 1e-3 | 1e-4 | 1e-4 |
| Hidden layer size for the MLP | 32, 64, 128, 256, 512 | 256 | 256 |

Table 7: Hyperparameters for adaptation module in relational policy adaptation

| Hyperparameter | Values tried | Best value | |
| --- | --- | --- | --- |
| | | Rearrangement | Navigation |
| Batch size | 8, 16, 32, 64 | 32 | 32 |
| Learning rate | 1e-3, 1e-4, 1e-5 | 1e-3 | 1e-3 |
| Dimension of the space in which the noun and the attribute of an entity are projected | 8, 16, 32 | 4 | 4 |
| Hidden layer size for the MLP used after the transformer module | 32. 64. 128. 256. 512 | 256 | 256 |
| No. of heads in the transformer module | 1, 2, 3, 4, 5, 6, 7, 8 | 4 | 4 |
| No. of encoder layers in the transformer module | 1, 2, 3, 4 | 2 | 2 |
| Dimension of the feedforward layer in the transformer module | 32, 64, 128. 256 | 128 | 128 |
| Dropout probability in the transformer module | 0.1, 0.2, 0.3, 0.4 | 0.1 | 0.1 |

