# OpenReview forum: "Language-guided Task Adaptation for Imitation Learning"
_TMLR — Withdrawn by Authors_

### Review · Reviewer_Bngk · 2023-01-10

**Summary Of Contributions:**

This paper explore an original task where the agent is provided a trajectory example, and another language instruction to define a new task goal (example, go to the other chair). Thus, the agent needs to attend to both modality to retrieve its final goal.
In addition, the author evaluates the task difficulty by training a goal prodictor, a relational policy adaptation and a final policy. They hence show the difficulty of the task, and how it may open new research direction.




**Audience:**

Yes

**Claims And Evidence:**

No

**Requested Changes:**

I would recommend the authors to
 - increase the test split
 - redo the full experimental protocol by:.
    - disentangling each game component (not just a final table with everything). For example, exploring goal prediction only as a standalone task, exploring different baselines and/or ablations. Then only focus on the policy, with goal prediction / oracle / traj+text / text-only / traj-only / no goal.
    - show training curves
    - add new metrics
    - explore common errors (with cherry-picked examples)
    - explore less naive training, e.g. using hindsight experience replay
 - only add natural language as a new standalone task (so far, everything is mixed)

Overall, the experimental section of the paper requires heay change before acceptance. On a general note, I would recommend the authors to think how they could highlight further the interest of their task through experiments.

**Strengths And Weaknesses:**

First of all, the paper is clearly written and quite enjoyable to read.
The task is interesting as it relates to generating new goals by combining reasoning with multiple modalities.

However, despite a good initial idea, the paper has core weaknesses:
  - While the task is interesting, and the dataset was well-defined, the paper did not introduce essential baselines. For instance, training the policy by conditioning and the text only, or the text+trajectory. Differently, the authors decided to train an independent goal generation model (I think this should be directly integrated into the agent input). However, they do not provide any metrics. On a different note, the interest in using negative-trajectory and language instruction is never truly shown. Finally, I would have liked additional metrics, e.g., agent convergence rate, average trajectory length, and normalized trajectory length (when an optimal trajectory exists). Overall, the paper test one model idea (that combines three models!) but never try to go for the simplest solution to really assess the underlying task
  - The test-set only uses 30 test trajectories. Please make it at least 10% of the data!)
  - The core of the paper is about reinforcement learning methods. However, there is no analysis of the literature. There is no reference to general value functions, nor goal relabelling (e.g., Hindsight experience replay), nor observational learning (e.g. Dagger). This lack of related work also appears in the final model as they do not rely on the modern classic RL approach to solve the task


Other remarks:
 - please do not use zero-shot terminology, as there the task is all about generalization within the goal space. There is no-zero shot aspect here (beware of language generality).
 - sweeping over the language model was weird. First, it is not the most important part of the paper (while being described as one of the first elements). Second. There is no provided ablation, and we do not know which model was used in the end.

---

### Review · Reviewer_vJ5H · 2023-01-11

**Summary Of Contributions:**

This paper proposes a solution to the zero-shot generalization of imitation learning, in which the task difference is explicitly given by the language modality.

[**Algorithmic contributions**] To achieve the adaptation, relational reward and policy adaptation are proposed to model how the agent estimates the (1) goal in target domains; (2) reward/distance functions; (3) entities relationship; and (4) policy adaptation.

[**Empirical contributions**] Furthermore, two novel benchmarks are given in the paper. The empirical evaluation of the proposed benchmarks indicates the effectiveness of each component of the proposed framework.

In general, I think the paper provides a simple and effective solution for the zero-shot generalization of imitation learning. Although the experimental design is simple and some parts can be further clarified, the work indeed gives a nice synthesis of the transfer/zero-shot imitation learning and language models for RL. In this initial review, I list the questions about the paper, and it is highly appreciated if the authors could respond and clarify them.

**Audience:**

Yes

**Broader Impact Concerns:**

No broader impact concerns

**Claims And Evidence:**

No

**Requested Changes:**

I believe the findings and solutions given in the paper are helpful for RL communities. However, some of the contents could in this version be further elaborated on or clarified. It would be highly appreciated if the authors could make some clarifications, and here I list some suggestions based on the proposed questions above.

**For Q1**: If time or computing resource permits, the authors could consider replacing the transformer with some of the computational models listed above, like GNN or deep set, which might give the agent a stronger inductive bias on relation modeling. Otherwise, discussing them and maybe clarifying why the transformer here is enough for modeling the relationships would also be acceptable.

**For Q2**: The authors could consider modeling the task with contextual and object-oriented MDPs. However, it is also OK to just discuss them in the paper as these are highly related models.

**For Q3**: If time or computing resource permits, the authors could evaluate these benchmarks. Otherwise, it is also acceptable to discuss them and explain why they are not suitable benchmarks for this work.

**For Q4-5**: Any discussion or clarification would be appreciated.

**For Q6**: Consider fixing some typos and adding a few contents. It is totally OK not to follow those writing suggestions (maybe with some justifiable reasons to explain why).


**Strengths And Weaknesses:**

### **Strengths**

(+) [**About the significance**] This work considers one of the most critical problems: zero-shot generalization in imitation learning, and provides with a simple solution via leveraging the direct language instructions. As I mentioned in the contributions section above, the problem itself is significant, and the solution idea is simple and effective.

(+) [**About the evaluation**] The proposed evaluation benchmarks are clear and could be useful for future research. And the empirical analysis in the paper is reasonable to show the effects of all components. In the appendix, the authors give all the hyper-parameter they tried (even though some of them might not be the optimal ones), this would be very helpful for the community to reproduce the work.

(+) [**About the presentation**] Though some parts could be further clarified, the overall presentation is clear and easy to follow.

### **Weakness and questions**
*Please note most of the points below are questions. Please feel free to correct me if I misunderstood anything. Though I may list a few related works, it is not a must to run all of them during the response phase, but any discussion would be highly appreciated.*

**Q1** [**About the relation modeling**] The work assumes that one can have access to the entity type and its attribute. I think it is reasonable to have such an assumption since we have many on-the-shelf algorithms to extract the entity-centric representation like the slot attention model. However, it seems that the relation modeling is done by using the transformer layer to learn the attention between entities as well as the encoded languages/goals. I am afraid this kind of implicit way of learning the relation may result in some redundancy. Why not the authors consider stronger inductive bias that may learn some explicit relational modeling among entities? For example, deep sets [1-2], GNN [3-4], relational networks [5-6], and explicit graph modeling [7].


**Q2**: [**About the MDP model**] I found the environment here may be better to be modeled or explained by contextual MDP [8] and OO-MDP [9]. In contextual MDP, there is a latent variable to be inferred in each task, similar to the changes on goal in this paper. Also, in [8], they use language instructions as surrogate models. In OO-MDP, the object is described with the attributes, and the relation/interaction among entities is included. It would be nice to give a more systematic model of MDP in this task.

**Q3**: [**About the evaluation**] I agree that the proposed two benchmarks would be very suitable to evaluate zero-shot imitation learning. However, it would be nice if the authors could show the effectiveness of this idea on more complex benchmarks that give real-world RL/IL scenarios. For example, the robotic manipulation task in [2] would be suitable for measuring it. [2] has provided the zero-shot imitation learning baselines already. The benchmark includes several objects and it is also doable to add the language instructions by directing using the techniques in this paper. Meanwhile, meta world [10] has its imitation learning baselines [11] and language contextual MDP baselines [8]. It might be another benchmark to use.

**Q4**: [**About the changes in dynamics**] Some zero-shot RL generalization also explore the effectiveness when the dynamics/state transitions changes across the source and target domains. I notice this work only considers the changes in the reward functions. Could this framework also generalize to the case where the dynamics also change?

**Q5**: [**About the policy adaptation**] In the paper, it is said that the action is executed in the environment when a goal-conditioned policy is learned and generated action. Is it only executed on the source domain?

**Q5**: [**Some writing suggestions**]
- Consider adding the section numbers.

- It would be better to have Figures 8 and 9 also in the main paper as they are important visualizations for the results.

- Consider adding another subsection on goal-conditioned RL in related works as it is related to the topic of this paper.

- In Table 2, making the highest (maybe highest non-oracle ones) bold would be helpful for readers.

- 1st line of the 2nd paragraph on Pag 3: missing the brackets for the reference.


**References**

[1] Zaheer, Manzil, et al. "Deep sets." NIPS 2017.

[2] Zhou, Allan, et al. "Policy Architectures for Compositional Generalization in Control." NeurIPS 2022 workshop on DeepRL.

[3] Li, Richard, et al. "Towards practical multi-object manipulation using relational reinforcement learning." ICRA 2020.

[4] Kossen, Jannik, et al. "Structured object-aware physics prediction for video modeling and planning." ICLR 2020

[5] Mambelli, Davide, et al. "Compositional Multi-Object Reinforcement Learning with Linear Relation Networks." ICLR workshop OSC 2022.

[6] Zambaldi, Vinicius, et al. "Deep reinforcement learning with relational inductive biases." ICLR 2018.

[7] Kansky, Ken, et al. "Schema networks: Zero-shot transfer with a generative causal model of intuitive physics." ICML 2017.

[8] Sodhani, Shagun, Amy Zhang, and Joelle Pineau. "Multi-task reinforcement learning with context-based representations." ICML 2021.

[9] Diuk, Carlos, Andre Cohen, and Michael L. Littman. "An object-oriented representation for efficient reinforcement learning." ICML 2008.

[10] Yu, Tianhe, et al. "Meta-world: A benchmark and evaluation for multi-task and meta reinforcement learning." CoRL 2020.

[11] Rivera, Corban G., et al. "Visual Goal-Directed Meta-Imitation Learning." CVPR 2022.

---

### Review · Reviewer_BkJH · 2023-01-12

**Summary Of Contributions:**

The paper proposes a transfer learning task where an RL agent is provided with a source demonstration, a linguistic description of an adaptation from that source, and a target demonstration. The two tasks are room rearrangement and room navigation, with 3 subtasks each.

The paper proposes two methods for initialization of a PPO agent, relational reward adaptation and relational policy adaptation, and claims they they can be used to transfer policies from source to target distributions and that a combined method using both improves the sample-efficiency of task adaptation compared to a PPO baseline.




**Audience:**

Yes

**Claims And Evidence:**

No

**Requested Changes:**

Required:
* Sample-efficient adaptation is arguably core to the paper - how do this relational policy adaptation perform when the goal-conditioned policy (step 1) is learned using only source demonstrations? Alternatively, quantify how many target samples relational policy adaptation uses and how this impacts Figure 7.
* Train until convergence
* Take another pass on the paper, with attention to the presentation order, style and structure as noted above. Use TMLR style file.

Strengthen:
* Strongly consider using these techniques on benchmark datasets (possibly could use Minigrid, for example)
* Consider minimizing experiments which are less relevant to the main point of the paper (i.e. language encoders and synthetic vs. natural language) to avoid confusion



**Strengths And Weaknesses:**

Strengths:
* The task itself is interesting and useful, especially for meta-learning (adapting decision-making policies via linguistic instructions). This can be used to extend RL techniques for instruction-following policies to instruction-encoded improvement/meta-learning policies
* The paper seems well-motivated in its limited setting, and explores relevant, orthogonal axes of variation of the problem
* The technique is generally interesting

Weaknesses:
* It seems likely that relational policy adaptation is transferring information about the target task during transformer finetuning, making this not necessarily a true zero-shot start for the adapted model (how do this technique perform when the goal-conditioned policy (step 1) is learned using only source demonstrations?)
* The natural language instructions are generated from templates, with a subset adapted by Mechanical Turk users (the number of which is unclear?), and represent a relatively small diversity of transfer learning tasks in real-world environments.
* Overall, the paper could use another pass for simplification of concepts and naming (for example, it's not clear that the framework is "relational" as stated, although the task data happens to require that, and the writing moves without separation between frameworks and concrete representations/implementations)
* One-hot entity encoding is unlikely to generalize to more complex settings (see Decision Transformer/Trajectory Transformer for approaches to generically encoding perceptions without requiring inductive biases)
* Training seems to have been halted before convergence? Which model has better asymptotic performance? How do you explain the combined model outperforming the oracle?
* Details on knowledge distillation are lacking
* Zero reward baseline is not clearly described, and may not be relevant
* Writing nits: Page 2: Adaption -> Adaptation
* Style nit: Should be a TMLR style file
* Writing is somewhat unclear when read linearly (requires multiple passes) - it may be helpful to simplify and compress the presentation. For example, Table 2 seems to present results from the Combined experiment on page 10
* Where is the PPO baseline in Table 2?

---

### Review · Reviewer_wrJb · 2023-01-17

**Summary Of Contributions:**

This paper introduces a setting that learns a target task without the demonstration for the specific task. Instead, only demonstration for a related source task and natural language description of difference between source task and target task are used. The authors propose two new datasets to support research in the proposed setting. They also propose a framework which includes two relational approaches for this setting.


**Audience:**

No

**Claims And Evidence:**

No

**Requested Changes:**

1. The motivation should be rethought.
2. It seems that the description of relational reward adaptation is more targeted at the Room Rearrangement task. Does the Room Navigation task also use $N_entities$ tokens to predict the location of each entity rather than just predict the agent’s location? Does the Room Navigation task also use $l_1$ norm distance rather than $l_2$ norm distance to predict the target goal state?
3. There is not enough explanation how the setting can be used for providing feedback to correct agent errors.
4. Are there any reasons that learning the potential function should be decomposed into two subproblems?
5. I’d like to see the performance of the learned goal-policy policy $\pi$ on the target task.
6. For the hyperparameters in table3, it seems that a larger model (number of encoder layers, hidden dimension) is always preferred. Why not increase the model size to see whether the performance can be further improved?


**Strengths And Weaknesses:**

Strengths:
1. The proposed setting is somewhat novel. It explores a new direction that combines NLP and RL and is well-contained.
2. The proposed datasets are different from existing datasets of related tasks.
3. The paper writing is fairly easy to follow, e.g., figures are well-designed to explain the ideas.

Weakness:
1. The motivation of this problem setting is not convincing. The introduction explains why it is challenging, but doesn't really justify why it is useful. Moreover, it doesn't justify why existing settings and benchmarks can not achieve the same goal. E.g., if you take a sequential decision making problem like navigating a maze, then the proposed source-target-adaptation problem is really like broken pieces of this navigation problem: at each step, the action of previous step is the source task, the description of the desired action is the language description, and the current step is the target task. Indeed, the examples in Figure-1 and Figure-2 are all like this. Additionally, in many existing navigation tasks, adaptation and generalization is really already there: e.g., in Chen and Mooney's maze navigation tasks, you train on 2 maps and test on 1 different map. So I am not sure what this new setting is really adding to the community.
2. The technical method is limited. There is not enough intuition to explain why the relational policy approach can work. You need to first learn a single policy function for both the source task and target task and later train the policy adaptation model to learn the policy generated from this model. What if the learned model itself is not good enough? Are there any evaluation results for the policy function in the first step? Can the model really infer the action for the target task only from the action for the source task and linguistic description?
3. There is no example for the demonstration of source task and target task for each benchmark. I don’t quite understand the difference between source task and target task, which seems to be different from the transfer learning that I know for the NLP area.
4. The success rate for relational policy adaptation on the Room Rearrangement task using natural language is quite low. The drop between natural language and synthetic language is significantly larger than the difference on the source task. Does it mean that relational policy adaptation relies on a powerful language encoder that recognizes the same entity/object when described differently? I notice that 4 kinds of language encoders are implemented and perform similarly. Does it mean that assistance from natural language is greatly limited?

---

### Note · Authors · 2023-01-31

I have read and agree with the venue's withdrawal policy on behalf of myself and my co-authors.